# Review of the Management of Obstructive Sleep Apnea and Pharmacological Symptom Management

**DOI:** 10.3390/medicina57111173

**Published:** 2021-10-28

**Authors:** Ladan Panahi, George Udeani, Steven Ho, Brett Knox, Jason Maille

**Affiliations:** 1Department of Pharmacy Practice, Texas A&M Rangel College of Pharmacy, 1010 W Ave B, Kingsville, TX 78363, USA; stevencuong0902@tamu.edu (S.H.); bknox@tamu.edu (B.K.); maille@tamu.edu (J.M.); 2Department of Pharmacy Practice, Texas A&M Rangel College of Pharmacy, 59 Reynolds Medical Building, College Station, TX 77843, USA

**Keywords:** obstructive sleep apnea (OSA), pharmacotherapy

## Abstract

Nearly a billion adults around the world are affected by a disease that is characterized by upper airway collapse while sleeping called obstructive sleep apnea or OSA. The progression and lasting effects of untreated OSA include an increased risk of diabetes mellitus, hypertension, stroke, and heart failure. There is often a decrease in quality-of-life scores and an increased rate of mortality in these patients. The most common and effective treatments for OSA include continuous positive airway pressure (CPAP), surgical treatment, behavior modification, changes in lifestyle, and mandibular advancement devices. There are currently no pharmacological options approved for the standard treatment of OSA. There are, however, some pharmacological treatments for daytime sleepiness caused by OSA. Identifying and treating obstructive sleep apnea early is important to reduce the risks of future complications.

## 1. Introduction and Background

Obstructive sleep apnea (OSA) is characterized by nighttime awakenings, hypoxia, and hypercapnia caused by the airway constricting partially or fully while sleeping [1]. The signs of OSA include daytime drowsiness, narcolepsy, cognitive impairment, and snoring while sleeping. Because these symptoms can go overlooked due to a relative lack of urgency, OSA often goes untreated. This increases the risk of conditions such as atrial fibrillation, stroke, heart failure, myocardial infarction, hypertension, and pulmonary hypertension [2,3,4,5,6]. An estimated forty to sixty percent of patients with cardiovascular disease (CVD) have OSA [7]. Many OSA patients also have diabetes mellitus, hyperlipidemia, and hypertriglyceridemia [8]. Overstimulation of the sympathetic nervous system, metabolic dysfunction, endothelial dysfunction, and systemic inflammation are all possible pathophysiological mechanisms that may be responsible for the risks of OSA [2,8].

## 2. Pathophysiology

### 2.1. Sympathetic Nervous System

Sleep is typically characterized by the dominance of parasympathetic activity in the body. Decreased oxygen and increased CO_2_ caused by an airway obstruction lead to increased sympathetic output in the periphery and central chemoreceptors [8,9]. Increased sympathetic output remains present during sleep and while awake. The study conducted by Narkiewicz et al. compared sympathetic responses among obese patients with or without OSA. The study showed that an increased sympathetic response was found in obese patients with OSA but was not found in the control obese group without OSA [9]. Because the obese patients without OSA did not have an increased sympathetic response, it is possible that OSA is the causative factor for having an increased sympathetic response [10]. The renin-angiotensin-aldosterone system (RAAS) is activated by the sympathetic neurons and, since the sympathetic response is found to be upregulated in patients with OSA, the RAAS system is also overstimulated. Patients with OSA often have elevated angiotensin II and aldosterone hormone levels in the body. Increases in these levels lead to water retention in the kidneys and vasoconstriction in the peripheral vasculature [11]. As a result of these mechanisms being activated, hypertension is commonly found in patients with OSA [12].

### 2.2. Endothelial Dysfunction

Endothelial cells normally release vasoactive and vasorelaxant factors to regulate the vascular tone. In OSA, the endothelial cells do not function in the same capacity [13]. Phillips et al., in a prospective study of OSA patients, measured oxygen saturation, hemodynamics, and changes in circulating endothelin-1 levels [14]. The study found that, after OSA treatment, patients experienced decreases in blood pressure and endothelin-1 [14,15]. Nitric oxide, which normally serves as a vasodilator, was found to have impaired action in OSA; however, the impaired action was reversible after CPAP treatment [16].

### 2.3. Systemic Inflammation

In OSA, there are increased inflammatory biomarkers such as interleukin-6 (IL-6) and tumor necrosis factor-alpha (TNF-a) [17,18,19]. This condition can be considered a low-grade chronic inflammatory disease [17,18,19]. In addition to the inflammatory biomarkers, concentrations of reactive oxygen species are increased due to the hypoxia caused by the night-time intermittent airway obstruction [20]. Increased reactive oxygen species in addition to the inflammation biomarkers mentioned above indicate a possible mechanism by which OSA increases risk of cardiovascular disease and overall mortality.

### 2.4. Metabolic Dysfunction

Type 2 diabetes mellitus (T2DM) is more prevalent in OSA populations. This is notable because T2DM may increase all-cause mortality and the risk of CVD. In a cross-sectional analysis of 2588 participants, the Sleep Heart Health Study found a link between obstructive sleep apnea and elevated fasting glucose, decreased glucose tolerance, and diabetes mellitus [21]. Nadeem at al. conducted a meta-regression analysis determining that there was increased LDL, triglycerides, and total cholesterol in patients with OSA [22]. This is notable because increases in these factors as well as blood glucose are risk factors for cardiovascular disease.

## 3. Epidemiology

Middle-aged and geriatric populations have a higher prevalence of obstructive sleep apnea globally. A literature-based analysis was conducted by Benjafield et al. in 16 different nations using the American Academy of Sleep Medicine (AASM) 2012 diagnostic criteria. In both men and women aged 30 to 69, there were roughly 936 million who had mild to severe OSA [23]. The analysis also found that nearly half of that, 425 million people, had either moderate or severe OSA [23]. The condition is still disproportionately undertreated and underdiagnosed despite such a high proportion of the population suffering from it. With a prevalence of one billion, estimates from one study show that 82% of men and 93% of women in the United States have obstructive sleep apnea but are undiagnosed [24,25]. The lack of focus on identifying and treating this condition is further emphasized by the strain it adds to the healthcare system. The estimated financial impact that undiagnosed OSA had on the United States in 2015 was $149.5 billion. Furthermore, $30 billion of that $149.5 billion is thought to be due to the increased risk of resulting cardiovascular and metabolic conditions and $86.9 billion from decreased productivity of individuals suffering from OSA. Notably, it was projected to have required less than one tenth of that total cost, $12.4 billion dollars, to diagnose and treat OSA that year [26,27].

## 4. Risk Factors

Risk factors that can cause OSA include obesity, gender, age, and genetic syndromes [25]. Screening patients for these risk factors and understanding the etiology of OSA is essential for beginning treatment to prevent economic burden and health risks.

### 4.1. Obesity

Obesity is a hallmark risk factor for the manifestation and disease progression of OSA [28]. Adipose is deposited around the circumference of the neck and airway of obese patients and may lead to increased risk of pharyngeal collapse. Airway collapse causes mechanical obstruction from fat buildup and is accompanied by loss of neural control that contributes to the development of OSA [29]. A study by Peppard et al. showed that there is a direct relationship between weight gain and OSA prevalence and apnea-hypopnea index (AHI), which is calculated as the number of episodes of obstructive apnea divided by the number of hours of sleep. A 10% increase in body weight increased the prevalence of moderate–severe OSA and AHI by six times [25,30]. Researchers at the Harvard T.H. Chaan School of Public Health speculate that one in two (48.9%) adults in the United States will become obese and that 25% will have severe obesity by the year 2030 [31]. It is predicted that the prevalence of OSA will increase as the prevalence of obesity increases in the years to come [32].

### 4.2. Gender

Men seem to be affected by OSA more than women. The SHIP-Trend study showed than men were at a greater baseline risk of obstructive sleep apnea than women by analyzing the prevalence of OSA in 1280 participants, with the result that OSA prevalence was 59% in the men studied compared to only 33% in the females [33]. Another study performed by Whittle et al. showed via MRI that men often have more fat deposition in the neck compared to women [34]. Men′s increased fat disposition causes increased neck circumferences and consequently puts them at a higher risk of airway collapse compared to women. Additionally, men′s airways tend to be longer than those of women. It is hypothesized that the extended airways put men at a higher risk of pharyngeal collapse [35]. Comparing individuals with the same BMI, OSA tends to be less prevalent in women compared to men [36]. The prevalence difference observed with gender is thought to be due to the role of sex hormones during the fertile age, which tend to disappear after menopause, and may influence the prevalence and severity of OSA in older females [37].

### 4.3. Age

Older-aged individuals are at a higher risk of having OSA. The previously mentioned SHIP-Trend study found that aging steadily increased the prevalence of AHI in men and women beginning at the age of 50 [33]. The suspected mechanism of how age influences OSA prevalence is from decreased genioglossus reflex to negative pressure, which impairs dilator muscle′s ability to compensate from pharyngeal collapse. Increases in type 1 collagen lead to delayed contractile-relaxant response in the pharyngeal constrictor muscle [38,39]. Because this compensatory response is decreased and the level of type 1 collagen in the pharyngeal constrictor muscle is increased, the contraction and relaxation response that is supposed to occur with each inspiration and expiration while sleeping is delayed [38,39]. The United Nations′ 2019 report on World Population Ageing completed by the Department of Economic and Social Affairs estimates that the proportion of the population over the age of 65 will increase from roughly 9% in 2019 to about 16.7% by the year 2050. This increase in elderly population, in tandem with the expected increase in obesity, is expected to result in an increased prevalence of OSA [40].

## 5. Signs and Symptoms

The patient may be largely unaware that they are exhibiting signs or symptoms of obstructive sleep apnea since they occur while the patient is asleep. Ascertaining a history from a spouse or partner of the patient is vital to completing a workup of their condition. The patient may be aware that they snore, or perhaps that they wake up gasping for air; however, their partner will likely be more aware of the frequency and severity of these signs. Frequent complaints from the spouses or patients with OSA include drowsiness, headaches upon waking, xerostomia, sore throat, and unrestful sleep. Daytime sleepiness or fatigue despite sufficient opportunities to sleep that is unexplained by other medical problems are other symptoms that can occur in OSA. Patients with OSA frequently present with a recessed mandible, a high Mallampati score, a high BMI, and a limited pharyngeal space [1,7,25]. Screening for obstructive sleep apnea consists of using either the yes or no questions in the STOP-BANG questionnaire or the Berlin questionnaire [1,8,25]. STOP-BANG questions include yes or no questions about the patient′s symptoms of drowsiness, absence of breathing during sleep, presence of hypertension, BMI more than 35 kg/m^2^, older than 50 years of age, neck circumference greater than forty centimeters, and male gender. If the screening produces a result indicating the presence of OSA, there is a recommendation for a sleep study utilizing polysomnography either at the patient′s home or in a lab [25].

## 6. Management and Treatment of OSA

### 6.1. Behavioral and Lifestyle Changes

Weight loss is the most important goal for overweight patients with obstructive sleep apnea because there is a direct correlation between increased neck fat disposition and OSA onset and its progression. The American Academy of Sleep Medicine (AASM) recommends a target BMI of 25 kg/m^2^ for overweight patients [41]. In 2019, a randomized controlled trial was conducted to analyze the effectiveness of an intensive weight-loss regimen and its effect on decreasing OSA severity in patients with severe OSA. They concluded that weight loss decreased OSA symptom severity and, additionally, these patients saw decreased cholesterol, biomarkers of inflammation, and blood glucose [42]. Positional therapy, where OSA patients sleep in a position other than on their back, has been found to be useful in preventing airway collapse. Sleeping in a supine position decreases inspirational volume, creates an unfavorable airway geometry, and limits functionality of muscles that dilate the airway [43]. AASM recommends the use of a pillow or backpack to prevent patients from sleeping in a supine position [41].

### 6.2. Continuous Positive Airway Pressure (CPAP)

Continuous Positive Airway Pressure (CPAP) is the first line intervention in OSA treatment. The mechanism behind this treatment is that it acts as a pneumatic splint by providing constant positive pressure into the airway through a mask interface, while still allowing for regular respirations [44]. The benefits seen from CPAP include increased quality of life, decreased daytime drowsiness, and decreased blood pressure [45,46,47]. AASM recommends CPAP as the first line agent in individuals with mild to moderate–severe OSA [41]. Unfortunately, as demonstrated by the Sleep Apnea Cardiovascular Endpoints (SAVE) clinical trial, CPAP improves OSA symptoms in mild and moderately severe cases but does not improve cardiovascular outcomes. Although there was significantly less daytime drowsiness and fewer instances of impaired breathing while asleep, there was not a difference in cardiovascular outcomes between the treatment and control groups consisting of 2717 participants with either moderate or severe OSA studied over 3.7 years [48].

### 6.3. Mandibular Advancement Devices or MAD

In patients who do not tolerate CPAP, another treatment option for OSA is a mandibular advancement device (MAD). These devices change the position of the lower jaw to move it forward and prevent the closing of the upper airway [49]. In both a systematic review and a meta-analysis of studies on CPAP and MAD, both CPAP and MAD decreased blood pressure. There was no significant difference between the two treatment options in blood pressure outcomes [50]. The AASM and the Academy of Dental Sleep Medicine (AADSM) created a joint practice guideline in 2015 that recommends that a mandibular advancement device customized to each person be used in patients with OSA instead of no therapy [51].

### 6.4. Surgical Treatment

There are two mainstay surgical options for OSA. One of the surgical treatments used is maxillomandibular advancement (MMA), which involves enlarging the airway by altering the position of the maxilla and the mandible. Another treatment called uvulopalatopharyngoplasty (UPPP) widens the airway by excising a section of the soft palate, uvula, and tonsil. The AASM recommends that patients be assessed for eligibility and contraindicated risk factors of surgical treatment before making decisions about these methods of therapy [41]. Ultimately, these treatments are reserved for those who do not respond well to CPAP or MAD. The authors of the SAMS trial analyzed how surgery affected the AHI and the Epworth Sleepiness Scale scores and compared the results to standard care. After six months, preliminary data from the trial concluded that patients with mild to moderate OSA improved with relation to apnea or hypoxia during sleep after either of the surgical methods described above [52].

### 6.5. Hypoglossal Nerve Stimulation

A hypoglossal nerve stimulator is similar to a pacemaker because it sends electrical impulses to the hypoglossal nerve during the breathing process, which stimulates the tongue to move away from the airway and prevent airway collapse [53,54]. A study, named the STAR trial, analyzed how hypoglossal nerve stimulation affected outcomes of obstructive sleep apnea. The trial recorded AHI, patients′ levels of drowsiness, sleep quality, prevalence of snoring, and additional polysomnography measures. Follow-up after three years showed a decrease in AHI events from a median of 28.2 events/h to 6.2 events/h [55]. Patients should be screened to see if they are an ideal candidate for hypoglossal nerve stimulation and preferably have moderate to severe obstructive sleep apnea, a BMI of 32, and a lack of complete centric collapse at the soft palate. Assessment of the location of collapse should be done with drug-induced sleep endoscopy before consideration for hypoglossal nerve stimulation [56,57].

## 7. Pharmacological Therapy for OSA

At this time there are no known pharmacological agents that are universally used, or FDA approved, for the management of OSA. Currently, FDA approved pharmacological agents are used strictly for symptom management, not for disease management. There have been trials studying the effectiveness of serotonergic drugs such as fluoxetine, paroxetine, trazodone, and mirtazapine; however, none of these agents have proved efficacious in improving disease severity or management [58]. Pharmacotherapeutic options for OSA and central sleep apnea (CSA) are limited, therefore many patients remain untreated [59]. The role of pharmacotherapy in OSA at this current time is limited to assisting in management of OSA associated symptoms and disease.

### 7.1. CNS Stimulants

Despite the efficacy of CPAP in the management of OSA, non-adherence can range from 46–83% [60]. However, even adherent patients can still experience residual daytime sleepiness while using CPAP [61]. Patients with daytime sleepiness due to OSA have a two-fold increased risk of occupational accidents, and driver sleepiness has been identified as one of the major causes of highway accidents and fatal crashes [62,63]. CNS stimulants such as modafinil and armodafinil can promote daytime wakefulness in this population [64]. Although the exact mechanism of action is still not fully understood, modafinil is thought to improve wakefulness by inhibiting dopamine and norepinephrine reuptake [65,66]. Interestingly, modafinil’s wake-promoting effects, unlike those of amphetamine, were not antagonized by the dopamine receptor antagonist haloperidol [67]. Therefore, modafinil and armodafinil have a unique profile with fewer adverse effects compared to those reported in traditional psychostimulants such as amphetamine or cocaine [65,66]. There are controversies regarding the impact of CNS stimulants on cognitive functions such as driving, since they can mask signs that a patient is physically in need of sleep [68]. Whereas modafinil is a mix of both the (S) and (R) enantiomers, armodafinil contains only the (R)-enantiomer and is thought to display slightly more pharmacological potency and longer duration of action in terms of wakefulness compared to modafinil [66]. Solriamfetol binds to the dopamine and norepinephrine transporters with low affinity and inhibits the reuptake of dopamine and norepinephrine with low potency [69]. Modafinil, armodafinil, and solriamfetol have FDA approval for excessive sleepiness associated with OSA [67,69,70].

One concern with CNS stimulant use is the potential for abuse associated with this drug class. Like other CNS stimulants, modafinil and armodafinil both produce psychoactive and euphoric effects with noted drug diversion and misuse during the armodafinil post-marketing period [67,70]. Moreover, modafinil compared to methylphenidate in terms of its ability to produce psychoactive and euphoric effects and feelings whereas solriamfetol showed abuse potential similar to or lower than phentermine [67,69,70]. Patients with a significant history of past abuse of other CNS stimulants such as methylphenidate, amphetamine, or cocaine should be followed more closely when taking modafinil, armodafinil, or solriamfetol [67,69,70].

Physical dependence is another potential limitation of CNS stimulant use. Abrupt cessation or dose reduction following chronic use of armodafinil can result in withdrawal symptoms, with the most common presentation being shaking, sweating, chills, nausea, vomiting, confusion, aggression, and atrial fibrillation [67]. In addition, abrupt withdrawal of armodafinil has caused deterioration of psychiatric symptoms such as depression [67]. Modafanil discontinuation has not been associated with specific withdrawal symptoms; however, sleepiness and fatigue are expected to return with discontinuation of use [70]. Similar to modafinil, there is no current evidence that abrupt discontinuation of solriamfetol will result in a consistent pattern of adverse events in individuals suggesting physical dependence or withdrawal [69]. Indeed, a risk/benefit discussion between physician and doctor should help determine the patient’s needs before prescribing a CNS stimulant.

### 7.2. Efficacy

Modafinil has been shown to improve residual daytime sleepiness in patients on CPAP [71,72]. After CPAP withdrawals, modafinil may help reduce sleepiness [73]. Some limited data are available on modafinil′s use in patients who are not using conventional treatments [74]. A trial showed that modafinil improved wakefulness in patients with mild to moderate OSA [74]. The most common side effects of modafinil were headache and nervousness [72]. Armodafinil has also been shown to help reduce excessive sleepiness in patients using CPAP [75,76]. The newest CNS stimulant, solriamfetol, helped increase wakefulness in patients with OSA [77,78]. Solriamfetol′s efficacy seems to persist regardless of adherence to conventional OSA therapy [79]. Although no direct head to head trials have been done to date, a recent indirect treatment comparison meta-analysis of a 12 week duration of the stated CNS stimulants led to varying levels of improvement in excessive daytime sleepiness with similar safety risks [80]. Dosing for daytime sleepiness and special considerations are listed in Table 1. 

### 7.3. Leukotriene Antagonists

For children who are experiencing labored breathing due to a sleep disorder, montelukast has been studied for its efficacy. Tonsillar and adenoid hypertrophy is a known risk factor for OSA, especially in the pediatric population [83]. The idea of how montelukast, a leukotriene antagonist, works in OSA is through reducing inflammation in the tonsils by inhibiting LT1 and LT2 receptors [84]. It is thought that montelukast influences the upper airway diameter [85]. Goldbart et al. studied the effects of montelukast on pediatric patients with OSA and found that there was a significant reduction in the respiratory disturbance index (RDI) over 16 weeks [86]. Kheirandish et al. studied the effects of combining budesonide nasal spray with montelukast in pediatrics and found that the therapy improved RDI after tonsillectomy or adenoidectomy [87]. Although preliminary studies are promising, there still is limited evidence available for the use of montelukast in pediatric patients with OSA and more research needs to be done to establish its role in therapy [86].

### 7.4. Inhaled Nasal Corticosteroids

Allergic rhinitis may exacerbate OSA by blocking the airway. It is believed that nasal corticosteroids can increase the upper airway diameter, alleviating symptoms of OSA [85]. Kiely et al. investigated the impact of nasal fluticasone on the AHI. The study had 23 participants and individuals with AHI values above 10/h were investigated deeper. These 13 individuals had a clinically significant reduction in AHI events from 30 to 23, a 27% reduction in AHI events. In the entire study population, there was a smaller statistically significant reduction from about 26 AHI events to 23, a reduction of only 15%. Using fluticasone also helped to improve daytime wakefulness, reduce nasal congestion, and improve air′s ability to move through the respiratory system [88].

### 7.5. Carbonic Anhydrase Inhibitor

Acetazolamide works by preventing the breakdown of carbonic acid, which leads to the accumulation of carbonic acid in the body and acidifies the pH of the blood. Due to the accumulation of carbonic acid, the kidneys secrete sodium, bicarbonate, chloride, and water in the urine. The clinical result is decreased blood pressure and metabolic acidosis [85,89]. Since acetazolamide increases blood pCO_2_, it is proposed that there is an interaction with hypoxic and hypercapnic stimulation and ventilation control through altering a process called loop gain [89,90]. Acetazolamide was shown to lower AHI in patients with OSA in many of the trials performed [85,89]. There was, however, not a clinically significant improvement in symptoms such as daytime sleepiness. Side effects were common in this population such as paresthesia and nocturia [85,89,91].

### 7.6. Other Pharmacological Therapies

Many pharmacological treatments have been studied in OSA but have varying efficacy. Gaisl et al. performed a meta-analysis on 58 clinical trials. The trials that showed statistically significant results over placebo consist of ondansetron with fluoxetine, spironolactone, spironolactone with furosemide, phentermine, zonisamide, dronabinol, liraglutide, and tramazoline [85]. Each of these trials had study populations of 40 or less except for liraglutide with 359 participants. These small study populations may be seen as limitations and the external validity of these studies questioned. Also, the clinical end points mostly studied were AHI; however, some other clinical endpoints studied were Epworth Sleepiness Scale (ESS) and maintenance of wakefulness test (MWT) [92,93,94]. It is important to consider what was studied: the baseline characteristics of the study population, the impact on AHI, and the side effect profile of these agents before using these agents off-label.

Many other agents such as clonidine, adalimumab, flumazenil, salmeterol, cilazapril, naloxone, pioglitazone, temazepam, flumazenil, salmeterol, cilazapril, naloxone, and pioglitazone, to name a few, were studied and showed statistically and clinically insignificant results [85]. Future studies may prove these agents to be clinically significant at a later time; however, at the time of publishing, these agents have not proven to have a place in therapy for OSA.

## 8. Conclusions

Obstructive sleep apnea has a high prevalence worldwide and is underdiagnosed and treated. There are many risk factors that contribute to OSA. This condition can increase one’s risk of metabolic and cardiovascular diseases, decreases quality of life, and increase mortality. Over the next few decades, when the prevalence of obesity and the elderly increases, the amount of people with OSA is expected to rise. For this condition to be better diagnosed and treated, healthcare providers must be better familiarized with the epidemiology and pathophysiology of OSA. Only after proper screening can the strain on the healthcare system caused by OSA and resulting comorbidities be reduced by proper treatment. Pharmacological options are scarce and current FDA approved drugs are only for symptom management. While pharmacological therapies have been studied, none are universally used for the treatment of OSA. Current first line therapies for the management of OSA consist of weight loss, CPAP, MAD, and surgical interventions.

## Figures and Tables

**Table 1 medicina-57-01173-t001:** CNS stimulant dosing and special considerations for daytime sleepiness.

Drug	Dose	Special Considerations
Armodafanil	150 mg to 250 mg PO once daily [81]	Use lowest effective dose in elderly to avoid potential toxicity Take in morning to avoid sleep interference Maximum daily dose is 250 mg
Modafanil	200 mg PO once daily [70]	Use lowest effective dose in elderly to avoid potential toxicity Take in morning to avoid sleep interference Maximum daily dose is 400 mg
Solriamfetol	37.5 mg PO once daily [82]	Titrate by double the dose in at least 3-day intervals Take in morning to avoid sleep interference Avoid use in patients taking Monoamine Oxidase Inhibitors (MAOIs) Maximum daily dose is 150 mg

## Data Availability

Not applicable.

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
