# Peer review of "Review of the Management of Obstructive Sleep Apnea and Pharmacological Symptom Management"

_medicina, 2021, doi:10.3390/medicina57111173_

Round 1

Reviewer 1 Report

The authors provide a very timely and needed review of the pharmacological treatment options for symptom management in the context of the overall  pathophysiology of OSA. The article will be a nice stepping stone for future research and education in this area. 

Author Response

Comments and Suggestions for Authors

The authors provide a very timely and needed review of the pharmacological treatment options for symptom management in the context of the overall  pathophysiology of OSA. The article will be a nice stepping stone for future research and education in this area.

Our Response

Thank you. No action or change needed. 

Reviewer 2 Report

Nice work and appropriate. CNS stimulants should be commented on the risk of dependence and abusis.

Author Response

Comments and Suggestions for Authors

Nice work and appropriate. CNS stimulants should be commented on the risk of dependence and abusis.

Our Response

A section was added for dependence and abuse for the CNS depressants. 

One concern with CNS stimulant use is the potential for abuse associated with this drug class. Like other CNS stimulants, modafinil and armodafinil both produce psychoactive and euphoric effects with noted drug diversion and misuse during the armodafinil post marketing period. 67, 70 Moreover, modafinil compared to methylphenidate in terms of its ability to produce psychoactive and euphoric effects and feelings whereas solriamfetol showed abuse potential similar to or lower than phentermine.67, 69, 70 Patients with a significant history of past abuse of other CNS stimulants such as  methylphenidate, amphetamine, or cocaine should be followed more closely when taking modafinil, armodafinil, or solriamfetol.69-71

Physical dependence is another potential limitation of CNS stimulant use. Abrupt cessation or dose reduction following chronic use of armodafinil can result in withdrawal symptoms, with the most common presentation being shaking, sweating, chills, nausea, vomiting, confusion, aggression, and atrial fibrillation.67 In addition, abrupt withdrawal of armodafinil has caused deterioration of psychiatric symptoms such as depression. 67 Modafanil discontinuation has not been associated with specific withdrawal symptoms, but sleepiness and fatigue are expected to return with discontinuation of use.70 Similar to modafinil, there is no current evidence that abrupt discontinuation of solriamfetol will result in a consistent pattern of adverse events in individuals suggesting physical dependence or withdrawal.72 Indeed, a risk/benefit discussion between physician and doctor should help determine the patient’s needs before prescribing a CNS stimulant.

Reviewer 3 Report

Change of the title might be considered since it is misleading. One might expect pharmacological cure for OSA, and only pharmacological treatment of OSA daily symptoms is described. Suggestion: "Review of the Management of Obstructive Sleep Apnea and Pharmacological Management of daytime symptoms" 

Author Response

Author Comment and Suggestions:

Change of the title might be considered since it is misleading. One might expect pharmacological cure for OSA, and only pharmacological treatment of OSA daily symptoms is described. Suggestion: "Review of the Management of Obstructive Sleep Apnea and Pharmacological Management of daytime symptoms" 

Our Response.

The title has been changed to the following:

Review of the Management of Obstructive Sleep Apnea and Pharmacological Symptom Management